# The Effect of Nanoparticle Additives on the Lubricity of Diesel and Biodiesel Fuels

**Vida Jokubynienė** [ID]**, Stasys Slavinskas** [ID] **and Raimondas Kreivaitis \***[ID]

Agriculture Academy, Vytautas Magnus University, K. Donelaičio Str. 58, 44248 Kaunas, Lithuania;
vida.jokubyniene@vdu.lt (V.J.); stasys.slavinskas@vdu.lt (S.S.)
\* Correspondence: raimondas.kreivaitis@vdu.lt

**Abstract:** Fuel lubricity is an essential property that ensures the longevity end efficiency of diesel CI engines. Nanomaterials have been shown to have the potential to improve lubricity in many different lubricating substances, including fuels. Moreover, the combustion process has also been shown to improve with the introduction of nanomaterials. This study investigated a series of nanoparticles, including carbon nanoplates, carbon nanotubes, aluminum oxide, zinc oxide, and cerium oxide, as lubricity-enhancing additives for selected fuels. Conventional diesel fuel and rapeseed oil methyl ester, referred to as biodiesel, were chosen as base fuels for modification. The lubricity was evaluated according to the standard test method ASTM 6079 using the HFRR tribometer. The leading lubricity indicators were the wear scar diameter, wear volume, and coefficient of friction. In addition, the worn surface analysis was performed to elucidate the lubrication mechanism. The results show that the addition of nanoparticles can improve the lubricity of both investigated fuels. However, the effect differed among nanoparticles and fuels. In summary, carbon nanotubes could be a rational choice for both fuels. In addition, zinc oxide improved the lubricity of diesel fuel, while carbon nanoplatelets and aluminum oxide nanoparticles showed improvements in the lubricity of biodiesel.

**Keywords:** nanoparticles; diesel fuel; rapeseed oil methyl ester; additives; lubricity; friction; wear

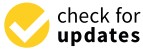



## 1. Introduction

Convenience, reliability, durability, and energy efficiency are essential areas in the automotive industry. Therefore, new technologies are being developed, including using less harmful materials, control of fuel combustion, and adequate lubrication, which helps to take a responsible view of environmental protection issues and efficient engine operation mode [1].

The number of cars worldwide reached 1.446 billion in 2022—a number that is increasing yearly [2]. Most vehicles are still powered by internal combustion (IC) engines, among which 29.9% are diesel engines [3]. The main disadvantage of IC engines is the harmful emissions of the fuels, which are nitrogen oxides (NOx), smoke, particulate matter (PM), carbon monoxide (CO), and unburned hydrocarbons (HC) [4]. Various studies are being carried out to reduce harmful emissions: (i) control of the combustion process [5], (ii) additional fuel cleaning or treatment technologies [6], and (iii) the use of catalysts and diesel particulate filters (DPF) [7]. In addition, alternative fuels are suggested [8]. However, using alternative fuels in diesel engines can have several consequences, including increased fuel consumption and NOx emissions, reduced engine power, piston ring sticking, and even complicated cold starting [9]. These disadvantages can be mitigated by proper combustion management and the use of functional additives. In recent years, more and more research has been done on nanomaterials, which are becoming promising diesel engine fuel additives [10,11].

The most commonly used nanomaterials are aluminum oxide ($Al_2O_3$), cerium dioxide ($CeO_2$), copper oxide ($CuO$), iron oxide ($\gamma$-$Fe_2O_3$), titanium oxide ($TiO_2$), zinc oxide ($ZnO$) nanoparticles (NPs), and carbon nanotubes (CNTs) [12–16]. Nanomaterials have a

high specific surface area and thermal capacity [17]. Morever, nanomaterials have unique physical, chemical, and thermal properties, allowing nanomaterials to be mixed in different fuels, leading to better engine performance and lower fuel emissions [14,16,18–21].

Prabu [22] compared additive-free biodiesel with a NPs loaded fuel mixture containing 20% biodiesel + 80% diesel + 30 ppm of $Al_2O_3$ + 30 ppm of $CeO_2$ in direct injection diesel engines. As a result, the engine thermal efficiency improved by 12%, NO emissions decreased by 30%, carbon monoxide decreased by 60%, hydrocarbon emissions decreased by 44%, and smokiness decreased by 38%. Chen et al. [23] conducted a study which showed that aluminum and silicon oxide blends in fuel resulted in a 1.76 times reduction of HC emissions due to shorter ignition delay. However, the emission of NOx was increased due to a higher heat release rate and a narrower combustion period. Nanthagopal et al. [24] investigated the effect of zinc oxide and titanium dioxide NPs in Calophyllum Inophyllum methyl ester (CIME) on exhaust emissions. Compared to neat diesel and additive-free CIME fuel, significant smoke emission reductions were observed for CIME-nano emulsions. Ooi et al. [25] investigated the performance of graphite oxide, single-walled carbon nanotubes, and cerium oxide NPs dispersed in diesel fuel. Based on their results, the ignition delay and combustion were shortened by 10.3% and 14.6%, respectively. Moreover, CO emission was reduced by 23.4%.

An equally important fuel property is lubrication. The fuel must lubricate the fuel pump, nozzles and other mechanisms of the injection system, thus reducing the friction and wear of rubbing surfaces. As a result, the longevity and energy consumption depend on fuel lubricity. The concern about the lubricity of fuels arose with the introduction of low-sulphur diesel fuel [26]. This sparked considerable interest, and led to many papers being published on the topic of diesel fuel lubricity. As a result, various fuel additives were suggested and employed. It was also revealed that introducing biodiesel could improve the lubricity of the final mixture [27]. With the rise of interest in nanomaterials, it was found that nanoparticles can be used as fuel additives to enhance lubrication [21]. Lai et al. [28] investigated the influence of NPs on the lubricity of fuels. It was observed that the addition of 0.1 wt. % of $Al_2O_3$ NPs significantly improved friction and wear. However, the authors did not reveal the underlying lubrication mechanisms. Xu et al. [20] investigated the tribological properties of nano-$La_2O_3$ additive in bio-oil formulated as diesel fuel. They have found that both wear and friction can be improved by using the optimal concentration of NPs. The positive effect was attributed to the adsorption of oil molecules onto the contact surfaces and the bearing effect of the NPs. The authors pointed out that at higher loads, adhesion takes place, while higher sliding speeds result in the abrasion of interacting surfaces. In a review paper, Kegl et al. [18] outlined a lack of studies on the tribological behavior of nanoparticles in diesel-fueled engines. Based on the available data, introducing NPs into the fuel could result in viscosity changes and lead to wear of the fuel injection system.

Based on the literature reviewed, there is a clear growing interest in application of nanomaterials in diesel fuels. In most cases, the investigation of the effect of nanomaterials is limited to the performance and emissions of IC engines. However, there are only a few papers describing the effect of nanoparticles on the lubricity of fuels. Therefore, this study aims to investigate the lubricity of a series of nanoparticles as potential additives for diesel fuel and biodiesel.

## 2. Materials and Methods

### 2.1. Materials Used

In this study, conventional diesel fuel (D), meeting the requirements of EN 590:2014+AC, and rapeseed oil fatty acid methyl ester biodiesel (RME), meeting the requirements of EN 14214, were used as base fuels (BF). The diesel fuel was obtained from "Neste" (Mažeikiai, Lithuania), while the biodiesel was obtained from "Rapsoila" Ltd (Mažeikiai, Lithuania). Both fuels were used as received, without any additional preparation. The main physico-chemical properties of the investigated fuels provided by the supplier are listed in Table 1.

**Table 1.** Properties of additive-free investigated fuels.

| Property | Standard | Value |
|---|---|---|
| Diesel Fuel | | |
| Kinematic viscosity @ 40 °C, cSt | EN ISO 3140 | 2.88 |
| Density @ 15 °C, kg/m$^3$ | EN ISO 3675 | 835 |
| Cetane number | EN ISO 4264 | 55.5 |
| Sulphur, mg/kg | EN ISO 20846 | 6.5 |
| Content of polycyclic aromatic carbohydrates, % | EN 12916 | 1.1 |
| Biodiesel | | |
| Kinematic viscosity @ 40 °C, cSt | EN ISO 3140 | 4.47 |
| Density @ 15 °C, kg/m$^3$ | EN ISO 3675 | 883 |
| Cetane number | EN ISO 5156 | 54.3 |
| Sulphur, mg/kg | EN ISO 20846 | <3 |
| Acid number, mg KOH/g | EN 14104 | 0.16 |
| Amount of monoglycerides, % | EN 14105 | 0.2 |
| Amount of diglycerides, % | EN 14105 | 0.13 |
| Amount of triglycerides, % | EN 14105 | 0.14 |

Several nanoparticles were selected to use as lubricity-enhancing additives. Carbon nanoplates (CPL), carbon nanotubes (CNT), aluminum oxide ($Al_2O_3$), zinc oxide (ZnO), and cerium oxide ($CeO_2$) were purchased from Sigma Aldrich (Burlington, MA, USA) and used as received. CPL, CNT, ZnO, and $CeO_2$ nanomaterials are powder-like, while $Al_2O_3$ nanoparticles are a 20 wt. % solution in isopropanol. The main characteristics of the investigated nanomaterials are presented in Table 2. Sorbitanmonooleat ethoxyliert (SPAN80) obtained in Sigma Aldrich was used to stabilize the dispersions of nanoparticles in fuels.

**Table 2.** Main characteristics of the investigated nanoparticles.

| Nanoparticles | Carbon Nanoplatelets, CPL | Carbon Nanotube, CNT | Aluminum Oxide, $Al_2O_3$ | Zink Oxide, ZnO | Cerium (IV) Oxide, $CeO_2$ |
|---|---|---|---|---|---|
| CAS No. | 308063-67-4 | 308068-56-6 | 1344-28-1 | 1314-13-2 | 1306-38-3 |
| Appearance | Pyrolytically stripped platelets (conical) >98% carbon basis. D × L 100 nm × 20–200 mum | Multi-walled >90% carbon basis. D × L 110–170 nm × 5–9 mum | Dispersion <50 nm, 20 wt. % in isopropanol | Nanopowder <100 nm particle size | Nanopowde <25 nm particle size |
| Density, g/cm$^3$ | 1.9 | 1.7~2.1 | 3.97 | 5.60 | 7.13 |
| Specific surface area (m$^2$/g) | 54 | 12.8 | 150 | 10–25 | 30–60 |
| Melting point, °C | 3652–3697 | 3652–3697 | 2045 | 1975 | 2600 |

### 2.2. Preparation of Fuel Samples

A total of 150 ppm (by wt.) concentration of nanoparticles in both fuels was selected. The powder-like nanoparticles were weighted accordingly, while the amount of $Al_2O_3$ solution was increased to get the same concentration of nanoparticles. A total of 150 ppm (by wt.) of SPAN80 was added to the mixture in order to stabilize the dispersions. The amount of surfactant was selected equal to the amount of nanoparticles. This ratio was selected based on studies reported in the literature [12,18].

The magnetic stirrer and ultrasonication were performed to prepare nanoparticle-loaded fuel samples. The preparation scheme is shown in Figure 1. First, the weighted mixtures were stirred on a magnetic stirrer at 20 °C and 3000 min$^{-1}$ for 20 min. Second, bath ultrasonication was performed at 60 °C for 30 min.

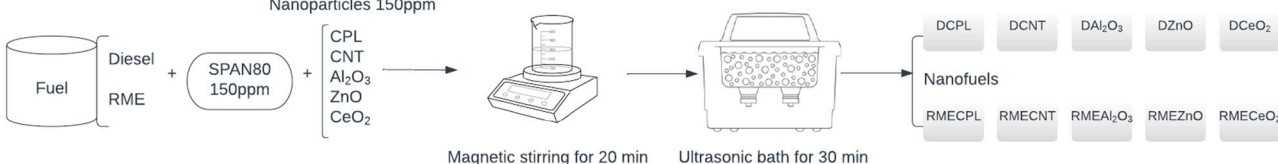

**Figure 1.** The flow scheme of nanoparticle-loaded fuels preparation.

The visual appearance of the prepared fuel samples is presented in Figure 2. The carbon nanoplatelets and nanotubes containing fuels are opaque black. With the introduction of other investigated nanoparticles, the color of the fuels did not change. Only the opacity of the $CeO_2$ nanoparticle-containing fuels increases. The prepared nanodispersions were stable for at least 5 h, which is enough to perform tribological experiments.

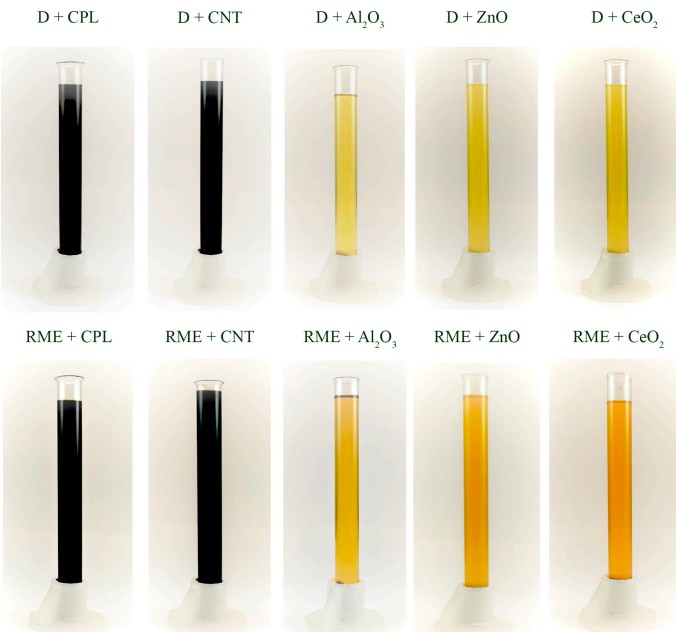

**Figure 2.** The visual appearance of the prepared nanoparticle-loaded fuels.

### 2.3. Physical Properties

The kinematic viscosities and densities of investigated nanoparticle-loaded fuels were determined using Anton Paar Stabinger viscometer SVM 3000 (Anton Paar, Graz, Austria). Three measurements were performed for each sample. As can be seen in Table 3, there was no significant difference between nanoparticle-free and nanoparticle-loaded fuels. Therefore, it was decided that the concentration of nanoparticles and surfactant was too small to change the viscosity or density of the base fuel.

**Table 3.** Physical properties of NPs loaded fuel samples.

| Fuel Sample | Density, 15 °C, g/cm³ | Kinematic Viscosity, 40 °C, cSt |
|---|---|---|
| Diesel Fuel | | |
| BF | 835 | 2.85 |
| CPL | 824 | 2.89 |
| CNT | 831 | 2.90 |
| $Al_2O_3$ | 841 | 2.81 |
| ZnO | 843 | 2.75 |
| $CeO_2$ | 840 | 2.82 |

**Table 3.** *Cont.*

| Fuel Sample | Density, 15 °C, g/cm$^3$ | Kinematic Viscosity, 40 °C, cSt |
|---|---|---|
| Biodiesel | | |
| BF | 885 | 4.49 |
| CPL | 882 | 4.58 |
| CNT | 879 | 4.51 |
| Al$_2$O$_3$ | 880 | 4.41 |
| ZnO | 887 | 4.50 |
| CeO$_2$ | 891 | 4.39 |

*2.4. Lubricity Evaluation*

The High-Frequency Reciprocating Rig (HFRR) Ducom TR-282 tribometer (Ducom, Karnataka, India) was employed to investigate the lubricity of fuel samples according to the standard test method ASTM 6079. The test conditions are listed in Table 4. During this test, the 6 mm ball is made to rub against the plate with a 1 mm stroke at a frequency of 50 Hz. The ball and the plate are made of bearing steel E-52100. The ball has a hardness of 750–800 HV30 and a surface roughness of 0.05 μm. The plate has a hardness of 190–200 HV30 and a surface roughness of 0.02 μm. After the test, both the ball and the plate are removed and cleaned. The wear scar diameter (WSD) of the ball is measured using an optical microscope, Nikon ECLIPSE MA 100 (Nikon, Tokyo, Japan). The cross-section profile of the wear trace on the plate is measured using a Mahr GD-25 stylus profilometer (Mahr GmbH, Goettingen, Germany). The measurements were performed perpendicular to the sliding direction in a few places along its length. The average cross-section area is multiplied by the length of the wear trace giving the wear volume. The worn surfaces were also inspected using Hitachi 3400N scanning electron microscope (SEM) (Hitachi, Tokyo, Japan). In addition, the worn surface composition on the plate was inspected using Bruker Quad 5040 energy dispersive spectroscopy (EDS) (Bruker, Berlin, Germany). The EDS was measured using a voltage of 15 kV and magnification of 2000.

**Table 4.** HFRR tribo-test conditions.

| Parameter | Units | Value |
|---|---|---|
| The volume of the fuel | mL | 2 |
| Stroke length | mm | 1 |
| Frequency | Hz | 50 |
| Test temperature | °C | 60 |
| Load | g | 200 |
| Test duration | min | 75 |
| Relative humidity | % | 42 |

During the tribo-test, the coefficient of friction (COF) was continuously recorded. Its variation as a function of test time is presented. Moreover, the mean values of COFs are calculated and presented in the manuscript. The coefficient of friction, the diameter of the wear scar on the ball and the wear volume were the main characteristics for lubricity evaluation. The proposed lubrication mechanisms were based on these results and worn surface analysis.

This study used the average of experiment repetitions, standard deviation, error bars, and least significant differences for statistical analysis.

**3. Results and Discussions**

*3.1. Friction Evaluation*

The coefficient of friction is proportional to the force required to move interacting surfaces. Therefore, the reduced friction force will save energy and improve mechanism

efficiency. In the case of diesel fuel, the reduced COF will result in less energy required for the fuel pump driving. The variation of the coefficient of friction observed in this study is presented in Figures 3 and 4, while its mean values are shown in Figure 5. The friction variation pattern differs for the two investigated kinds of fuels. Lubrication with biodiesel provides low COF, which has a similar value throughout the tribo-test. Diesel fuel, on the other hand, has a higher COF with a pronounced running-in period. This behavior could be related to the composition and viscosity of fuels [27,29,30]. According to the composition given in Table 1, the investigated biodiesel contains mono and diglycerides, which possess an adsorption ability. The adsorption of molecules is a self-initiating process and, in most cases, does not require additional energy. If the adsorption layer is strong enough, it provides low and stable friction. Moreover, no running-in is required. It was observed by Hu et al. [29] that even a tiny amount of monoglycerides presented in biodiesel could significantly improve lubricity. In the present study, biodiesel contained 0.2 wt. % of monoglycerides, which could result in low COF. The higher viscosity was also in favor of biodiesel (Table 1). The higher viscosity of lubricants can facilitate the separation of interacting surfaces and prevent direct contact.

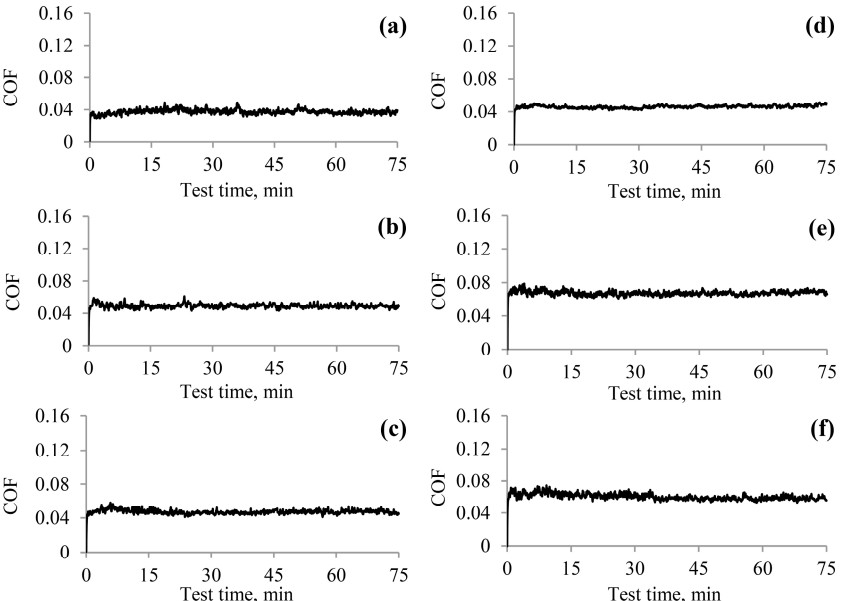

**Figure 3.** COF variation during the tribo-tests when additive-free biodiesel (**a**) and CPL (**b**), CNT (**c**), Al$_2$O$_3$ (**d**), ZnO (**e**), and CeO$_2$ (**f**) nanoparticle-loaded biodiesel samples were investigated.

The low-sulphur diesel fuel contains friction and wear-reducing additives to ensure its lubricity. Fully formulated diesel fuel meeting the standard requirements was used in the present study. The COF variation observed in the diesel fuel tribo-test represents a typical action mechanism of antiwear additives. The running-in period appears at the onset of the test when friction-induced tribo-reactions occur. The COF stabilizes when tribo-film is built. In the present case, it took 40 min to reach the steady state friction. The adsorbed layer of molecules can provide lower friction compared to antiwear additives. In this study, biodiesel possesses 1.8 times lower steady-state COF than diesel fuel.

Due to the different friction reduction mechanisms, the investigated fuels responded differently to NPs additives. Modifying biodiesel with NPs resulted in higher COF (Figure 5). However, the friction variation pattern did not change—there were no running-in periods, and COF kept the same value throughout the tribo-test (Figure 3). It must be noted that Al$_2$O$_3$ nanoparticle-loaded biodiesel has the least fluctuating COF among the investigated biodiesel samples (Figure 3d). It is the only positive factor among the nanoparticle-loaded biodiesel samples.

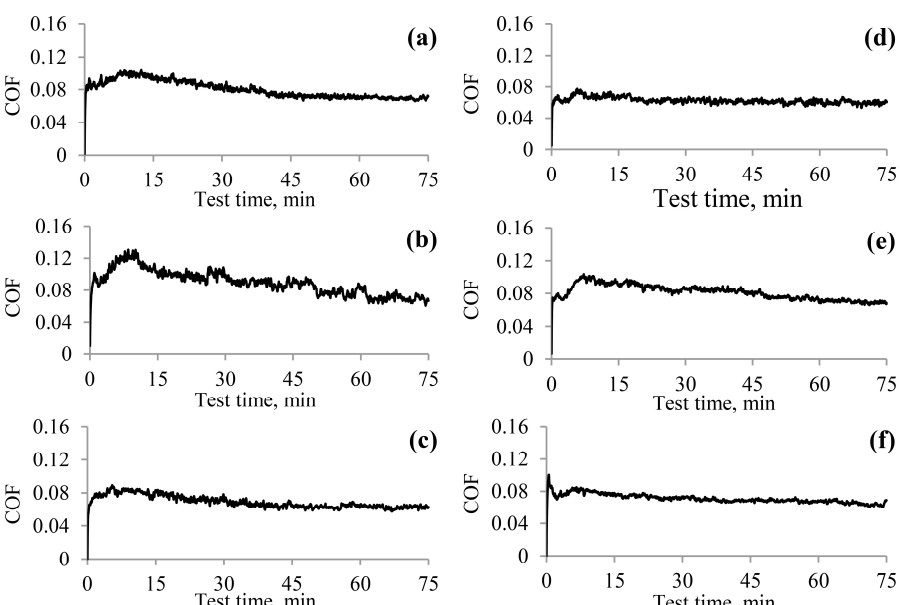

**Figure 4.** COF variation during the tribo-tests when additive-free diesel fuel (**a**) and CPL (**b**), CNT (**c**), $Al_2O_3$ (**d**), ZnO (**e**), and $CeO_2$ (**f**) nanoparticle-loaded diesel fuel samples were investigated.

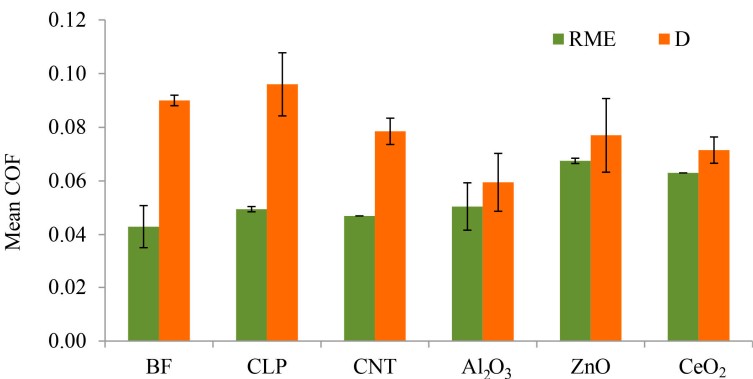

**Figure 5.** Mean coefficient of friction obtained in the tribo-tests.

Introducing nanoparticles in diesel fuel resulted in the changed friction variation pattern (Figure 4). Modification with ZnO NPs did not affect friction variation. CNT, $Al_2O_3$, and $CeO_2$ NPs provided diesel fuel with a shorter running-in period, resulting in lower mean COF. Moreover, the maximum COF observed during the running-in period was lower. The introduction of CPL NPs increased COF, extended the running-in period, and provided a more fluctuating coefficient of friction. The positive effect of nanoparticles in diesel fuel is a shorter running-in period and lower maximum and mean COFs values. It must be noted that all the diesel fuel samples show similar friction after approximately 60 min of testing (Figure 4). This may be because the additives present in diesel fuel govern steady state COF, while NPs provide a shorter running-in period.

According to the mean COF, the investigated fuel samples could be ranked as follow:

- Biodiesel $\leq$ CNT $\leq$ CPL < $Al_2O_3$ < $CeO_2$ < ZnO;
- $Al_2O_3$ < $CeO_2$ < CNT $\leq$ ZnO < diesel fuel < CPL.

### 3.2. Wear Evaluation

In order to ensure the longevity of fuel pumps and nozzles, diesel fuels must possess a particular wear reduction ability regulated by the standard EN 590. According to the standard, the wear scar diameter obtained in the HFRR tribo-test must be $\leq$460 μm.

Therefore, the smaller WSD is always preferred. In this study, both the WSD and wear volume were evaluated. The results are presented in Figure 6.

**Figure 6.** Wear volume (**a**) and wear scar diameter (**b**) obtained during the tribo-tests.

According to the WSD, all the investigated samples meet the requirements of the standards. As a result, the obtained wear scar diameters fit between 200 and 250 µm. On the other hand, the wear volume has considerably wider variation among the investigated fuel samples. Unfortunately, there was a high variation among the repetitions. According to the statistical analysis, the biodiesel samples must show a $34.3 \times 10^3$ µm$^3$ difference between samples to have a significant difference. The significant difference for diesel fuel samples is even higher—$50.9 \times 10^3$ µm$^3$.

In the HFRR tribo-test, the nanoparticle-free base fuels showed considerably different wear reduction abilities, whereas diesel fuel was superior to biodiesel. The introduction of nanoparticles improved the wear-reduction ability of biodiesel. In the case of diesel fuel, adding carbon nanotubes and zinc oxide nanoparticles has a marginally positive effect on wear-reduction ability. However, this improvement is not statistically reliable. Unfortunately, adding carbon nanoplatelets, aluminum oxide, or cerium oxide nanoparticles resulted in adverse effects, as more intense wear was observed. According to the wear volume, the investigated fuel samples could be ranked as follow:

- $Al_2O_3$ = ZnO = CNT < CPL < $CeO_2$ < biodiesel;
- CNT = ZnO < diesel fuel < $CeO_2$ < $Al_2O_3$ < CPL.

It must be noted that in some investigated cases, the wear reduction results are opposite to the results of friction reduction. For instance, introducing nanoparticles to RME improved its wear reduction while COF was increased. The contradiction between wear and friction occurs when friction energy coming into the system is released as frictional heat or surface deformation instead of wear [31,32]. The surface deformation leads to pushed-out material to the sides of the worn scars. If the material loss due to wear is calculated, this part is not considered as wear [33]. It was observed that only carbon nanotubes could provide a benefit for both fuels. The modification of RME with carbon nanotubes resulted in a 42% wear reduction, while the COF remains similar to the nanoparticle-free sample. On the other hand, carbon nanotubes provided diesel fuel with almost 8% wear reduction and 12% COF reduction. If, however, considering the effect of nanoparticles on biodiesel, it can benefit from modification with carbo nanoplatelets and aluminum oxide nanoparticles, and 32 and 44% wear reduction was observed. Unfortunately, the application of both nano additives resulted in slightly higher COF, while $Al_2O_3$-loaded fuel showed COF stabilization.

### 3.3. Analysis of the Worn Surfaces

The comprehensive analysis of the worn surfaces could reveal prevailing wear mechanisms and lubrication patterns. The optical images of the wear scar on the ball and wear

traces on the plate, and its profile, measured perpendicular to the sliding direction, are presented in Figures 7 and 8. In addition, the high-magnification SEM images of wear traces on the plate and the EDS composition of corresponding worn surfaces are presented in Figures 9 and 10. Finally, the roughness of the wear trace on the plate measured perpendicular to the sliding direction is listed in Table 5.

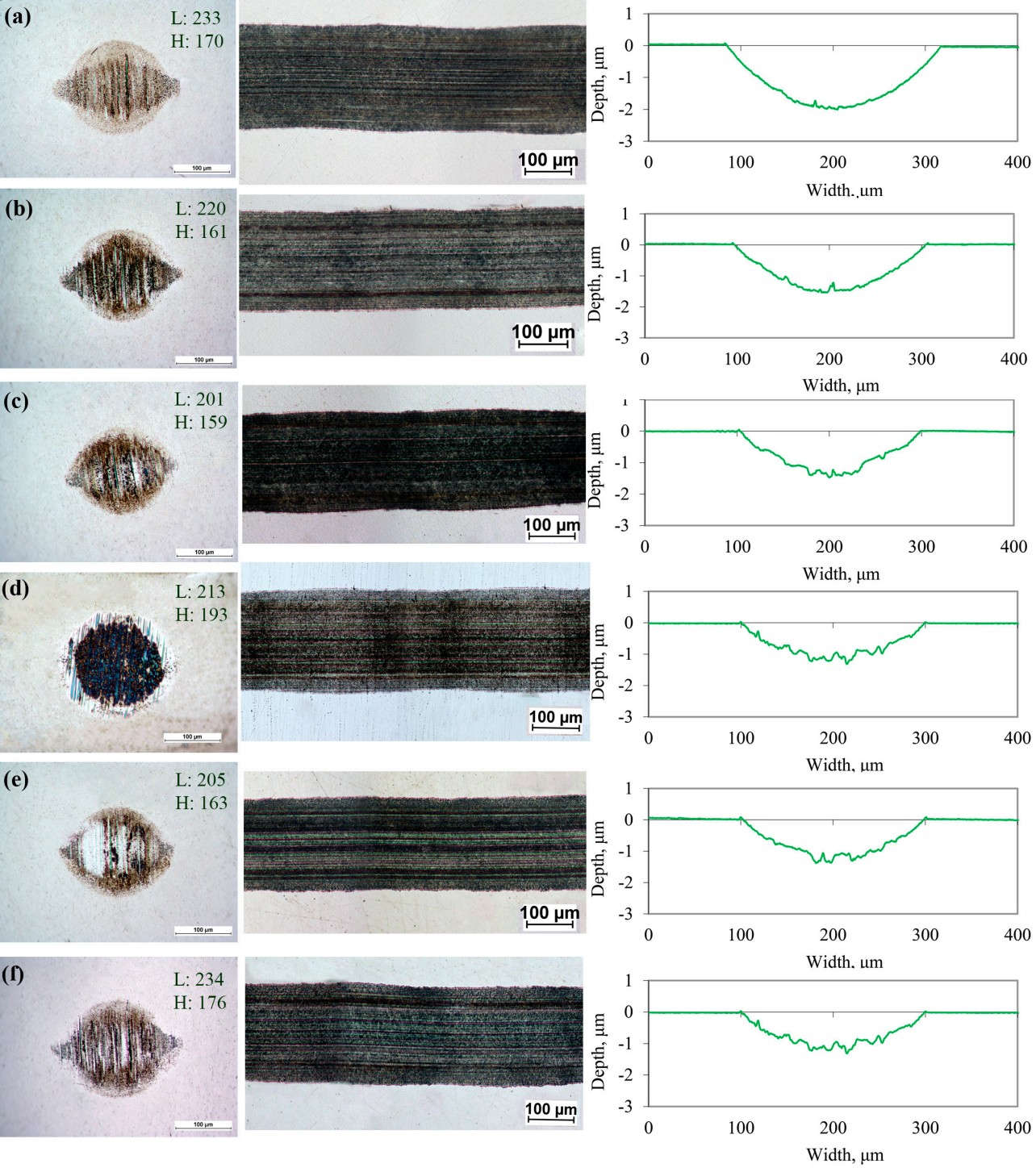

**Figure 7.** Optical microscope images of the wear scar on the ball, section of the wear trace on the plate, and profile of the wear trace (from left to right) observed after the tribo-test when nanoparticle-free biodiesel (**a**) and CPL (**b**), CNT (**c**), Al$_2$O$_3$ (**d**), ZnO (**e**), and CeO$_2$ (**f**) nanoparticle loaded biodiesel were used for lubrication.

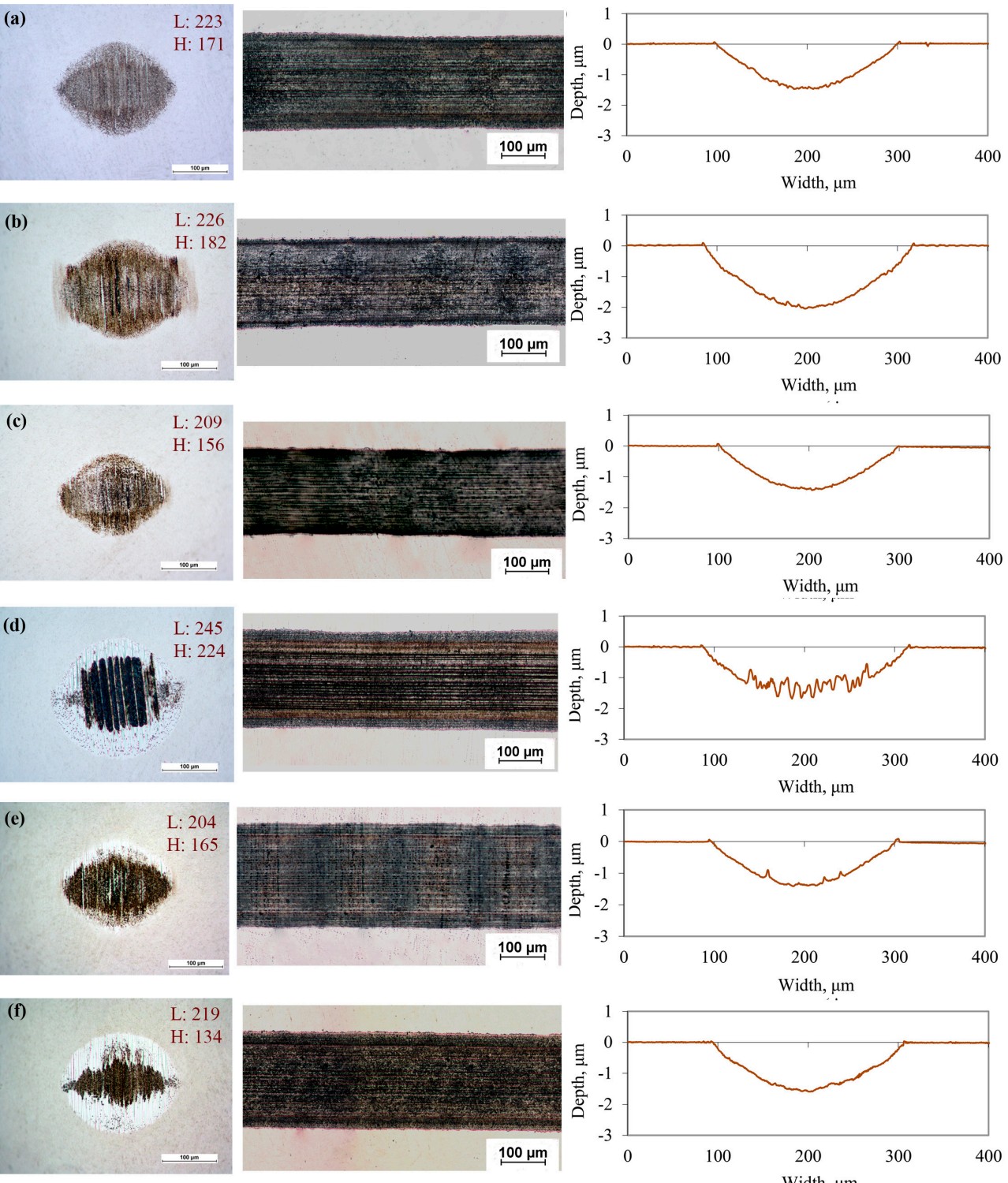

**Figure 8.** Optical microscope images of the wear scar on the ball, section of the wear trace on the plate, and profile of the wear trace (from left to right) observed after the tribo-test when nanoparticle-free diesel fuel (**a**) and CPL (**b**), CNT (**c**), $Al_2O_3$ (**d**), ZnO (**e**), and $CeO_2$ (**f**) nanoparticle loaded diesel fuel were used for lubrication.

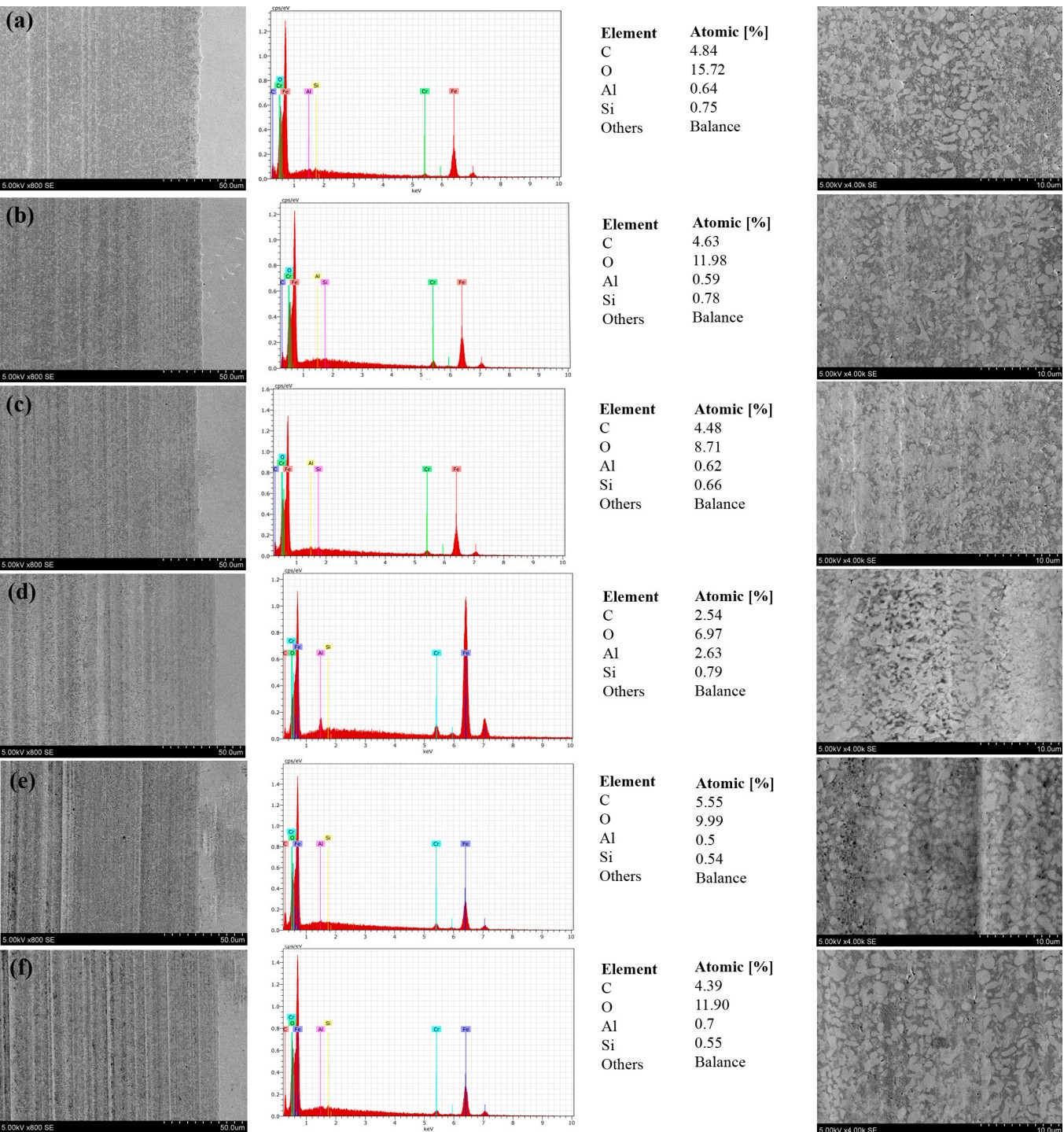

**Figure 9.** SEM images and EDS analysis results observed after the tribo-test when nanoparticle-free biodiesel (**a**) and CPL (**b**), CNT (**c**), Al$_2$O$_3$ (**d**), ZnO (**e**), and CeO$_2$ (**f**) nanoparticle-loaded biodiesel were used for lubrication.

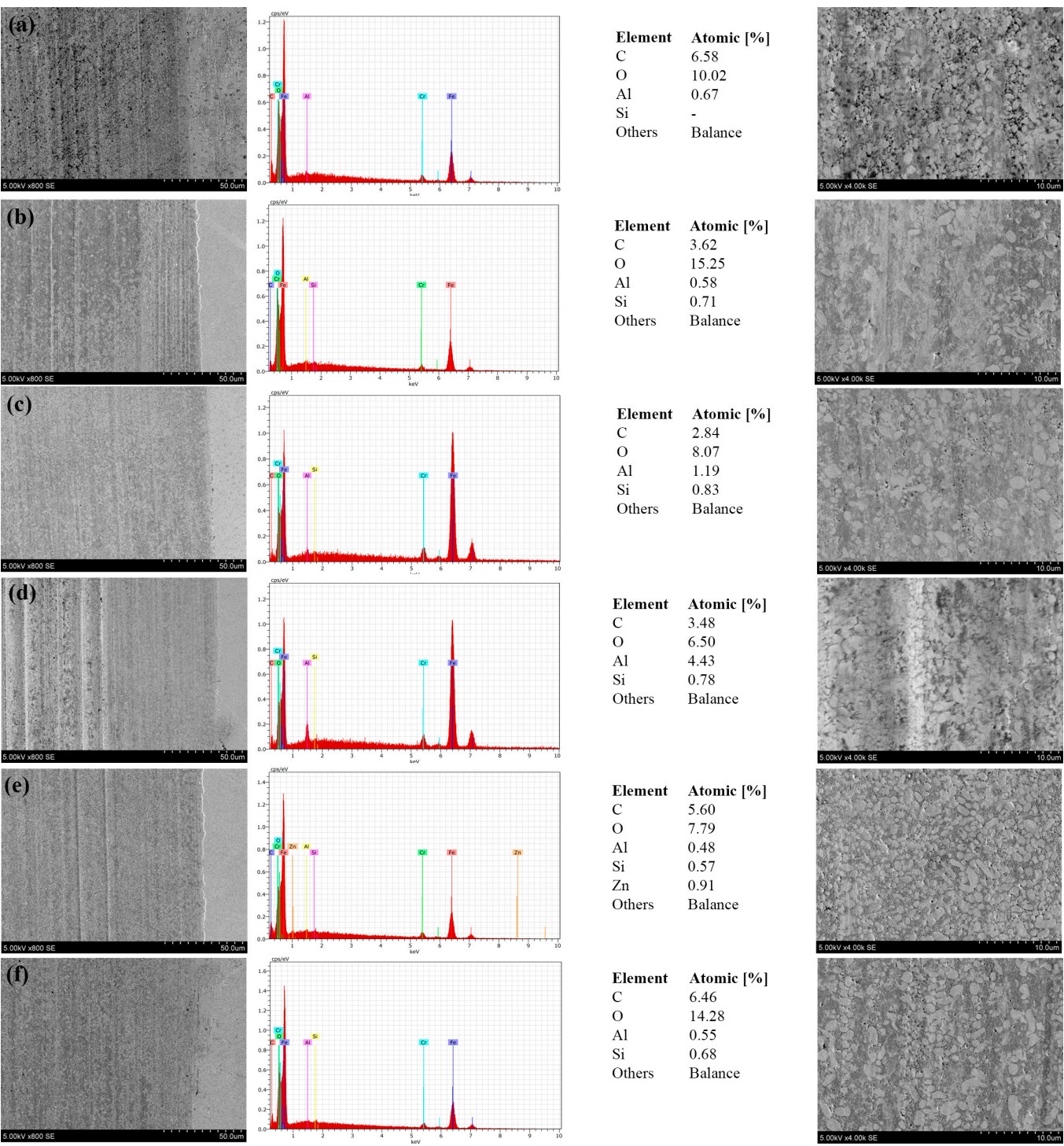

**Figure 10.** SEM images and EDS analysis results observed after the tribo-test when nanoparticle-free diesel fuel (**a**) and CPL (**b**), CNT (**c**), Al$_2$O$_3$ (**d**), ZnO (**e**), and CeO$_2$ (**f**) nanoparticle−loaded diesel fuel were used for lubrication.



**Table 5.** The roughness of the wear trace on the plate.

| Fuel Sample | Ra, μm | Rz, μm |
|---|---|---|
| Diesel Fuel | | |
| BF | 0.015 | 0.070 |
| CPL | 0.018 | 0.093 |
| CNT | 0.009 | 0.048 |
| $Al_2O_3$ | 0.088 | 0.397 |
| ZnO | 0.012 | 0.062 |
| $CeO_2$ | 0.011 | 0.051 |
| Biodiesel | | |
| BF | 0.012 | 0.084 |
| CPL | 0.020 | 0.102 |
| CNT | 0.026 | 0.127 |
| $Al_2O_3$ | 0.039 | 0.178 |
| ZnO | 0.024 | 0.111 |
| $CeO_2$ | 0.021 | 0.103 |

The observed wear scars reveal marginal wear of the balls. A residue of tribo-film, surface polishing, and minor scratches could be discerned in the optical images of these worn surfaces. On the other hand, the plate surface underwent much more intense wear with distinct features. Therefore, the above-discussed friction and wear results could be reflected on the worn surfaces.

As was mentioned earlier, lubrication with nanoparticle-free RME provided low friction, while adding nanoparticles resulted in higher COF. The reason can be seen by analyzing the wear traces on the plate in Figure 7. The lubrication with RME produced a relatively smooth worn surface (Table 5). However, it has a few tiny scratches, and a part of the worn material is pushed to the sides of the wear trace (Figure 9a). With the introduction of nanoparticles, furrows were formed in the wear trace. As a result, the roughness of the worn surfaces increased (Table 5). Moreover, scratches were also observed on the much harder ball surface (Figure 7). It could be the case that the three-body wear occurred when nanoparticles were present between interacting surfaces. The surface deformation required more energy to keep surfaces in relative motion. Therefore, higher COF was observed, and none of the nanoparticles reduced friction when used as additives in RME.

Nanoparticles could also be embedded in the softer plate surface and cause abrasion of the ball. The composition of the worn surface shows that a higher amount of aluminum was found after lubrication with aluminum oxide nanoparticles loaded biodiesel (Figure 9d). There was no evidence of other nanoparticles in the wear trace after corresponding tribo-tests. However, their amount could be too small to be detected with EDS. The reason why nanoparticles provide wear reduction while increasing COF is unclear. It is proposed that the rolling of nanoparticles and their agglomerates occur during surface interaction. Therefore, the hard particles cause the deformation of the interacting surface while having low wear. On the other hand, low COF and higher wear in the case of lubrication with nanoparticle-free RME could result from tribo-corrosion. The acidic components presented in vegetable-origin esters can cause corrosive wear. The nanoparticle additives were found to reduce this negative effect [20].

The nanoparticle-free and nanoparticle-loaded diesel fuel-lubricated surfaces were also relatively smooth (Figure 8). The introduction of nanoparticles resulted in surface polishing. The sample modified with AlO nanoparticles was an exception. Very rough worn surfaces were produced on both the plate and the ball. Interestingly, it resulted in lower COF compared to the nanoparticle-free diesel fuel. It could be that the rolling of nanoparticles occurs between interacting surfaces during the tribo-test, which results in lower COF. In the case of diesel fuel, aluminum oxide and zinc oxide were found in the wear trace on the plate.

According to the EDS composition measurements, all the worn surfaces contain high amounts of carbon and oxygen. It was observed that the amount of carbon was not higher when carbon-based nanomaterials were used as additives. Therefore, it was proposed that carbon came from the fuels. However, the amount of carbon does not correlate with wear or friction. On the other hand, a higher amount of oxygen was found in the wear traces, which underwent more intense wear.

## 4. Conclusions

In this study, the effect of nanoparticles on the lubricity of diesel fuel and biodiesel was experimentally investigated. The HFRR tribo-tests were performed to evaluate the lubricity. Based on the obtained results, the following conclusions could be made:

- The different natures of investigated fuels resulted in different lubricity responses when nanoparticles were introduced;
- Nanoparticle-free biodiesel possesses very low friction. Introducing nanoparticles resulted in higher friction. In contrast, nano additives improved the wear-reduction ability of biodiesel;
- Nanoparticle-free diesel fuel showed higher friction, which was reduced by adding nanoparticles. No wear improvement was observed in the case of diesel fuel modification with nanoparticles. It must be noted that the introduction of carbon nanoplatelets and aluminum oxide nanoparticles increased the wear;
- Considering the lubricity requirements of the EN 590 standard, any of the investigated nanoparticles could be used to modify fuels for a diesel-fueled IC engine.

**Author Contributions:** Conceptualization, R.K. and V.J.; methodology, R.K. and V.J.; software, R.K.; validation, R.K., S.S. and V.J.; formal analysis, R.K.; investigation, S.S. and V.J.; resources, R.K. and S.S.; data curation, V.J.; writing—original draft preparation, R.K. and V.J.; writing—review and editing, R.K.; visualization, V.J.; supervision, S.S.; project administration, R.K. and V.J.; funding acquisition, R.K. and V.J. All authors have read and agreed to the published version of the manuscript.

**Funding:** This research received no external funding.

**Institutional Review Board Statement:** Not applicable.

**Informed Consent Statement:** Not applicable.

**Data Availability Statement:** Not applicable.

**Conflicts of Interest:** The authors declare no conflict of interest.

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
