# Peer review of "The Effect of Nanoparticle Additives on the Lubricity of Diesel and Biodiesel Fuels"

_lubricants, doi:10.3390/lubricants11070290_

Round 1

Reviewer 1 Report

In this manuscript, the author selected two types of diesel fuel and studied the effect of nanoparticles on their anti-friction and anti-wear properties. However, they did not clarify why the same nanoparticles exhibit different tribological behaviors in the two types of fuels. Additionally, since the reduction of friction coefficient plays an important role in improving fuel economy, the primary function of additives is to achieve the goal of reducing friction coefficient. In addition, the nanoparticles discussed in this article cannot significantly lower the friction coefficient, especially for biodiesel. 

Based on the above judgment, I think that this article does not meet the standards for acceptance and publication.

Besides that, there are several problems and suggestions. Some comments are as follows:

1.     Line 159 “This behavior could be related to the composition and viscosity of fuels.” The author pronounces that the difference of friction coefficient between two kinds of diesel oil is caused by the difference of composition and viscosity. Line 167 “The higher viscosity of lubricants can facilitate separate interacting surfaces and prevent direct contact.” The impact of lubricant viscosity on the friction coefficient typically depends on the lubrication regime. It is oversimplified to assume that an increase in viscosity will always lead to a reduction in the friction coefficient.

2.     Line 288 “It is proposed that the rolling of nanoparticles and their agglomerates occur during surface interaction. Therefore, the hard particles cause the deformation of the interacting surface while having low wear.” Generally, the rolling of nanoparticles is beneficial to the reduction of COF, but in fact the experimental results are contradictory. The author proposes that hard particles cause deformation of the interacting surface while having low wear, but does not provide evidence or explanation to support the claim.

3.     Line 296: “It could be that the rolling of nanoparticles occurs between interacting surfaces during the tribo-test, which results in lower COF.” In the diesel system, the author mentions that the rolling of nanoparticles reduces the friction coefficient, which is obviously contradictory to the previous discussion.

4.     The authors should consider discussing the potential impact of adding SPAN80 to stabilize the dispersions on the coefficient of friction. If any relevant studies or data are available, it would be beneficial to include them in the article to provide a more complete understanding of the tribological properties of the system studied.

5.     The author should consider discussing the potential impact of adding alumina in the form of isopropyl alcohol solution to diesel fuel on its tribological properties, including any effects that isopropyl alcohol may have.

6.     The authors are advised to expand the list of references to provide more comprehensive support for their arguments and ideas.

There are some syntax error in the article and need the authors to revise it carefully.

Author Response

The authors are very grateful to the reviewer for the opportunity to improve this manuscript.

The reply to the reviewer is in a PDF file.

Author Response

(The authors gave the same response as above.)

Reviewer 3 Report

JokubynienÄ— et al. conducted an experimental investigation of the tribological performance of two diesel fuels from different sources, adding various nanoparticles to them. The article is well-structured, making it easy to follow. The test setup and sample selection adhere to established standards, and the lubricant preparation routines are well-described, which lends credibility to the results presented. That being said, there is some room for improvement, both in content and in the presentation of the results. I provide my suggestions for the content below. As for the presentation, I've provided my comments in the attached PDF file.

Content

- The motivation behind the additive selection appears to be unclear. In the introduction, the authors present these additives as commonly used to reduce harmful emissions. However, it remains unclear why they were selected for lubricity tests.

- A surfactant molecule was added to all of the mixtures in the same quantity as the nanoparticle molecules. This surfactant:
    - Can potentially serve as a friction modifier.
    - May interact differently with nanoparticles depending on the base solvent (type of diesel) and the type of nanoparticles used.
    - Could be actively involved in the formation of a surface protective film, and most likely will be.
These effects, however, are not discussed in the study.

- One of the additives, Al2O3, is used as a 20% blend with isopropanol. This essentially means that this sample, in addition to the nanoparticle additive and surfactant, is modified by isopropanol. Without delving into the discussion of how this could complicate the pool of molecular interactions in the mixture, it could noticeably affect the mixture's viscosity and density. For instance, see this reference: Lapuerta, M., et al. (2010). Energy & Fuels, 24(8), 4497-4502. DOI: 10.1021/ef100498u. However, the authors state that the effects on density and viscosity are negligible. While this may indeed be the case, it would necessitate a separate discussion, and providing the original data on viscosity and density would be beneficial for the reader. This is particularly important since the viscosity is later used to explain the differences in performance. Furthermore, it would be necessary to provide the rationale for using this sample as an isopropanol blend, in contrast to the other additives.

- Another question arises concerning the Al2O3 sample. Upon analyzing figures 4 and 5, it appears that blends with Al2O3 generally exhibited good performance. The biodiesel sample modified with Al2O3 ultimately showed the best performance in terms of wear and friction (a statistically significant result). Could this be related to the presence of isopropanol? If not, why did this particular sample perform the best?

- The authors do not address potential repeatability issues with the test results. It is unclear how many tests were performed for each lubricant mixture.

- The presentation of the wear data results is confusing, making it impossible to evaluate them.

- The conclusions drawn do not take into consideration the statistical significance of the results. See comments in the attached file.

- The ultimate result of the evaluation is the superior performance of the unmodified biodiesel oil. While the nature of this result is unclear, it seems valuable, given that this fuel comes from a renewable source and is the least modified in the series, thus having the smallest environmental footprint. It might be worth elaborating more on this result.

Presentation

See attached PDF file.

Author Response

(The authors gave the same response as above.)

Reviewer 4 Report

The paper is promising but needs extensive editing, reduction in length and re-consideration.

The 'Introduction' is too long and needs serious reduction/editing. The emphasis should be on the possible use of nano-materials in fuels to reduce energy and wear in fuel pumps. Some examples given are not directly compared. The current Introduction states the need for optimal ratio but this is not addressed in the subsequent parts of the paper. The case for choosing the nano-materials used is not made. 

Titles to Tables should be in 'Leading Caps'.

In 'Materials and Methods', Table 1 should be reduced in size as many of the fuel properties are in common and the differences are readily handled.

      What is the reason for choosing 150ppm of nanoparticles as their concentration?,  - not given, similarly the reason for choosing 150ppm SPAN80.

Table 3 - no reason for choosing the tribe-test conditions in Table 3 are not given.

Results and Discussions - line 160 - what is meant by ' - The fatty acid containing biodiesel possesses an adsorption ability'.

      In the discussion  of the effect of the nano-particles on the two fuels, it would be better if the effects are described separately for each fuel and discussed briefly before comparing them.

     I found the colours used for the results in Figure 3 to be confusing, some colours are quite close. Clearly, there is no positive effect for adding nano-materials to the bio-diesel. Further, there is only marginal improvement in adding nano-materials to the diesel fuel after the 'running -in' period of 75 minutes run duration. 

     Because of the substantial 'running-in' period for the diesel fuel, the use of 'mean coefficients of friction' at line 199 is meaningless.  The subsequent Figure 4 needs to be re-calculated on the basis of the results at 75 minutes rather than the 'average', because of the long period for stabilisation for the diesel tests. 

      I do not understand the reasoning of the first paragraph in S.3.2., line 204. In the second paragraph, surely if the WSD values are less than 300um, and the standard of 460um is reached. This must be clarified.

  Figure 6 should declare that it is for the bio-diesel fuel test at the beginning of the Fig.6 title, similarly for Figure 7.    

line 283 - ' - plate - ', instead of 'pate'. 

The authors should re-consider the discussion commencing at line 283.

The conclusion commencing at line 306 should stay more clearly that there is no advantage in adding the nano-particles tested to bio-diesel and its friction coefficient but with improved wear reduction. In addition the authors should consider if the small reduction in friction coefficient for diesel caused by addition of nano-particles is sufficient cause for using them when there is no wear reduction improvement.

     I cannot see any evidence for the second point, line 317,  made in Section 6, for the full term results.

     I trust that the authors will re-consider these points in re-submitting this paper.   

Needs gentle improvement 

Author Response

(The authors gave the same response as above.)

Round 2

Reviewer 1 Report

Although the author has made serious revisions to the article, but they has not provided targeted responses to the review comments, and the key scientific issues have not been addressed.  

Based on personal experience, I suggest that the author carefully consider the design of the experimental plan.

Author Response

Responses to the reviewer's comments were provided in the PDF file. In this case, we provide them in this field.

Author's comments on reviewers' remarks

Reviewer's comments

In this manuscript, the author selected two types of diesel fuel and studied the effect of nanoparticles on their anti-friction and anti-wear properties. However, they did not clarify why the same nanoparticles exhibit different tribological behaviors in the two types of fuels. Additionally, since the reduction of friction coefficient plays an important role in improving fuel economy, the primary function of additives is to achieve the goal of reducing friction coefficient. In addition, the nanoparticles discussed in this article cannot significantly lower the friction coefficient, especially for biodiesel. 

Author's response

The different behaviour of nanoparticles is still a question which requires many investigations to be answered thoroughly. In this case, we wanted to show the influence of nanoparticles, which are intensively investigated for combustion optimisation, on lubricity. Therefore a variety of nanomaterials was selected.

The COF reduction is significant from the energy point of view. However, the wear is even more critical because it will cause damage to the equipment. Consequently, even more, energy will be required. This paper presents the results of lubricity, which could be further used to compromise friction, longevity, and emissions.

Reviewer's comments

  1. Line 159 "This behavior could be related to the composition and viscosity of fuels." The author pronounces that the difference of friction coefficient between two kinds of diesel oil is caused by the difference of composition and viscosity. Line 167 "The higher viscosity of lubricants can facilitate separate interacting surfaces and prevent direct contact." The impact of lubricant viscosity on the friction coefficient typically depends on the lubrication regime. It is oversimplified to assume that an increase in viscosity will always lead to a reduction in the friction coefficient.

Author's response

We agree with the reviewer that lubricants' viscosity is not always the leading parameter. In the investigated case, the reciprocation motion leads to the boundary and mixed lubrication regimes. In these regimes, lubricants' viscosity is very important. The high reciprocation sliding speed of 50 Hz also can benefit from higher viscosity. Therefore viscosity was mentioned as one of the essential factors.

Reviewer's comments

  1. Line 288 "It is proposed that the rolling of nanoparticles and their agglomerates occur during surface interaction. Therefore, the hard particles cause the deformation of the interacting surface while having low wear." Generally, the rolling of nanoparticles is beneficial to the reduction of COF, but in fact the experimental results are contradictory. The author proposes that hard particles cause deformation of the interacting surface while having low wear, but does not provide evidence or explanation to support the claim.

Author's response

We agree with the reviewer that nanoparticle rolling is generally considered a friction reduction mechanism. However, the effect depends on the properties of interacting surfaces and nanoparticle/agglomerate size. According to Hanke et al. [Wear 267 (2009) 1319-1324], the energy which comes into the system and is recorded as friction energy is not always proportional to wear. In their study, the wear-friction relationship was related to particle size. They also pointed out that the environment and material properties are important.

Reviewer's comments

  1. Line 296: "It could be that the rolling of nanoparticles occurs between interacting surfaces during the tribo-test, which results in lower COF." In the diesel system, the author mentions that the rolling of nanoparticles reduces the friction coefficient, which is obviously contradictory to the previous discussion.

Author's response

We agree with the reviewer that nanoparticle rolling is generally considered a friction reduction mechanism. However, the effect depends on the properties of interacting surfaces and nanoparticle/agglomerate size. According to Hanke et al. [Wear 267 (2009) 1319-1324], the energy which comes into the system and is recorded as friction energy is not always proportional to wear. In their study, the wear-friction relationship was related to particle size. They also pointed out that the environment and material properties are important.

Reviewer's comments

  1. The authors should consider discussing the potential impact of adding SPAN80 to stabilize the dispersions on the coefficient of friction. If any relevant studies or data are available, it would be beneficial to include them in the article to provide a more complete understanding of the tribological properties of the system studied.

Author's response

SPAN80 was selected as a surfactant to stabilize nanoparticle dispersions. Several studies use SPAN80 to stabilize different nanoparticles in diesel and biodiesel fuels. Moreover, this surfactant was also used in lubricants [Kegl T., et al., Progress in Energy and Combustion Science 83 (2021) 100897].

In most studies, the amount of surfactant is selected according to the amount of nanoparticles. We have studied the stability of several surfactant concentrations, and 150 ppm was found to have the best stabilizing effect. Therefore in this study, 150 ppm of SPAN80 was used.

We do not want to outline the results of stability tests because it is a topic for another study.

Unfortunately, we did not investigate the effect of SPAN80 alone on the lubricity of fuels.

Reviewer's comments

  1. The author should consider discussing the potential impact of adding alumina in the form of isopropyl alcohol solution to diesel fuel on its tribological properties, including any effects that isopropyl alcohol may have.

Author's response

We agree with the reviewer that adding any counterpart will affect all the properties of the final solution. However, in the present case, there were 0.075 wt. % of the solution in which 0.015 wt. % were nanoparticles. The studies which focus on the influence of alcohol are investigating much higher concentrations. Based on that, we assume that such a small amount of isopropanol has a marginal effect on the lubricity.

Reviewer's comments

  1. The authors are advised to expand the list of references to provide more comprehensive support for their arguments and ideas.

Author's response

We have added several literature sources.

The authors are very grateful to the editor and reviewers for the opportunity to improve this manuscript.

Reviewer 4 Report

The paper is now clearer and easier to read. I now recommend that it is published.   

Author Response

The authors are very grateful to the reviewer for the opportunity to improve this manuscript.

Round 3

Reviewer 1 Report

The author carefully revised the article and answered the questions raised by the reviewers.

However,  there are still many problems with the use of nano additives in fuel, and there is still a long way to go from practical applications, so the authors should pay attention to the further research.